# Single nucleotide polymorphisms and copy-number variations in the *Trypanosoma brucei* repeat (TBR) sequence can be used to enhance amplification and genotyping of *Trypanozoon* strains

Nick Van Reet[1]*, Pati Patient Pyana[2], Sara Dehou[1], Nicolas Bebronne[1], Stijn Deborggraeve[1¤], Philippe Büscher[1]

1 Department of Biomedical Sciences, Institute of Tropical Medicine, Antwerpen, Belgium, 2 Département de Parasitologie, Institut National de Recherche Biomédicale, Kinshasa, Democratic Republic of the Congo

¤ Current address: Médecins Sans Frontières—Access Campaign, Geneva, Switzerland
* nvanreet@itg.be

## Abstract

The *Trypanosoma brucei* repeat (TBR) is a tandem repeat sequence present on the *Trypanozoon* minichromosomes. Here, we report that the TBR sequence is not as homogenous as previously believed. BLAST analysis of the available *T. brucei* genomes reveals various TBR sequences of 177 bp and 176 bp in length, which can be sorted into two TBR groups based on a few key single nucleotide polymorphisms. Conventional and quantitative PCR with primers matched to consensus sequences that target either TBR group show substantial copy-number variations in the TBR repertoire within a collection of 77 *Trypanozoon* strains. We developed the qTBR, a novel PCR consisting of three primers and two probes, to simultaneously amplify target sequences from each of the two TBR groups into one single qPCR reaction. This dual probe setup offers increased analytical sensitivity for the molecular detection of all Trypanozoon taxa, in particular for *T.b. gambiense* and *T. evansi*, when compared to existing TBR PCRs. By combining the qTBR with 18S rDNA amplification as an internal standard, the relative copy-number of each TBR target sequence can be calculated and plotted, allowing for further classification of strains into TBR genotypes associated with East, West or Central Africa. Thus, the qTBR takes advantage of the single-nucleotide polymorphisms and copy number variations in the TBR sequences to enhance amplification and genotyping of all Trypanozoon strains, making it a promising tool for prevalence studies of African trypanosomiasis in both humans and animals.

## Introduction

The subgenus *Trypanozoon* comprises various species, subspecies and subtypes of the unicellular protozoan *Trypanosoma brucei (T.b.)*, all causing diseases in humans or animals [1, 2]. *T.b.*

**Data Availability Statement:** All relevant data are within the manuscript and its Supporting Information files.

**Funding:** PB received grant OPP1174221 from The Bill and Melinda Gates foundation (gatesfoundation.org) and grant CHARHAT-RDC from the Departement Economie, Wetenschap & Innovatie (EWI-Vlaanderen.be). NVR received grant 1.5.093.16N from the Fonds Wetenschappelijk Onderzoek (fwo.be). The funders had no role in study design, data collection and analysis, decision to publish, or preparation of the manuscript.

**Competing interests:** The authors have declared that no competing interests exist.

*gambiense (T.b.g.)* is responsible for chronic human African trypanosomiasis (HAT), a disease targeted for elimination by the World Health Organization, that still accounted for 977 patients reported in West and Central Africa in 2018 [3]. Annually, less than 100 cases are due to *T.b. rhodesiense (T.b.r.)*, which causes acute HAT in East Africa [4]. *Trypanosoma brucei brucei (T.b.b.)* is a subspecies causing animal African trypanosomiasis (AAT) in an extensive range of game and domestic animals in Sub-Saharan Africa, but is considered non-human infective. All these *T.b.* subspecies are transmitted by tsetse flies (*Glossina* spp.), that also act as vectors for other subgenera such as *Dutonella* and *Nannomonas*, which are other causative agents of AAT [5]. The remaining species in *Trypanozoon*, *T. evansi (T. ev.)* and *T. equiperdum (T. eq.)*, cause non-tsetse transmitted African trypanosomoses (NTTAT). These parasites are dyskinetoplastic mutants of *T.b.* that have largely or completely lost their mitochondrial genome, and with it the ability to develop in and be transmitted by tsetse flies [6, 7]. Further genetic studies have shown that some of the species and subspecies can be further divided in types or groups that each have their own peculiarities [8–11]. In mammals infected with *Trypanozoon* parasites, severity and disease progression may vary depending on the genotype of the parasite and the host [12–14].

Molecular diagnosis of *Trypanozoon* infections often requires targeting multi-copy nucleic acid sequences to increase the chance of detecting the sparse parasites in blood and other tissues [15–18]. The molecular target with the highest known copy-number in *Trypanozoon* is the 177 bp long *Trypanosoma brucei* repeat (TBR) sequence. TBR sequences are direct tandem repeats that form the central core of the minichromosomes (MCs) and the few intermediate chromosomes present in the nucleus [19]. Their organization as a large repetitive palindrome, running from both subtelomeres to a central inversion point, indicate a role as origin of replication in these chromosome classes [19]. Around 100 MCs, sized 50–150 kb, are present in the nuclear DNA of *T. brucei* and they represent almost 10% of the nuclear genome [19]. It is estimated that roughly 55% of each MC, and thus 5.5% of the nuclear DNA in *T. brucei*, consists of such TBR repeats [19, 20]. The non-repetitive DNA on MCs carries an important part of the silent VSG gene repertoire, with most MCs having complete VSG genes that can be transposed to the VSG expression site during the early stages of an infection [20]. In *T.b.g.*, the average lengths of the MCs are smaller, being 25 to 50 kb, and the estimated copy-numbers vary between a few to up to 100 [21–24].

Soon after the discovery of the MCs as part of the African trypanosome satellite DNA [25, 26], the TBR sequence was chosen as target for diagnostic PCRs for screening *Trypanozoon* infections in mammals and insects [27, 28]. Over the past years, several other TBR PCR were developed for use in conventional and quantitative PCR [29–32]. Yet, despite suggestions by Sloof *et al.* [25] and others [20, 33, 34], TBR sequence heterogeneity in *Trypanozoon* was never extensively addressed. In this study, we provide evidence that TBR sequences are far more heterogenous than previously assumed. Furthermore, we show that single nucleotide polymorphisms and copy-number variations in the TBR sequences can be exploited to improve the amplification of all *Trypanozoon* taxa using a newly developed quantitative TBR-PCR, called qTBR, that may even allow to suggest the geographical origin of certain strains.

## Materials and methods

### *Trypanozoon* collection

Trypanosome sediments of *Trypanozoon* strains and cloned populations were available as DEAE purified pellets kept at -80°C [35–37] (S1 Table). They were prepared from *in vivo* expansions in mice or rats for which clearance was issued by the Animal Ethics Committee of the Institute of Tropical Medicine (DPU2017-1). DNA was extracted from 50 µl pellets,

corresponding to 5 x $10^7$ trypanosomes, using the Maxwell 16 Tissue DNA Purification Kit (Promega), eluted in 300 μl of elution buffer, aliquoted at 10 ng/μl and stored at -20˚C. Most populations were previously typed according to specific genetic markers for *T.b.g.* I [36], for *T. b.r.* [37], for *T. ev./eq.* A [35] and for *T. ev.* B [38]. Strains negative for these (sub)species specific markers, yet positive for M18S II [15], were considered either as *T.b.b.*, *T.b.g.* II or *T. eq*, depending on host, geographical origin or described genetic background [10, 39]. None of the *Trypanozoon* strains harboured mixed infections, as determined by testing the minisatellite marker MORF2-REP [40].

## BLAST search for TBR sequences

The TBR sequence [K00392.1] was queried in the Trypanosoma Blast Server using BLASTn on the Sanger Institute website [https://www.sanger.ac.uk/resources/software/blast/] against the *T.b.b.* EMBL data [/corebio/data/blastdb_web/tryppub/embl] and *T.b.g.* I "reads" database [/corebio/data/blastdb_web/tryppub/TBGAMBIENSE.reads] in February 2017. The first 100 hits for *T.b.b.* and *T.b.g.* that had query scores above 800 were reverse complemented or not, and aligned using MUSCLE in CLC SequenceViewer 8.0 (S1 File). Next, individual TBR sequences were extracted from each hit using the restriction site *HhaI* as start point (S1 File). Individual TBR sequences were realigned in MUSCLE and sorted according to the presence of a few key SNPs into TBR sequence sets according to subspecies and sequence size. This resulted in the construction of the Tbb177 and Tbbr176 TBR sequence sets from the *T.b.b.* database and the Tbg177 and Tbg176 TBR sequence sets from the *T.b.g.* database (S1 File). Consensus sequences for each TBR sequence set, representing 80% of the variants encountered, were aligned to the original TBR sequence and the 177-T1 and 177-T2 TBR variants described by Wickstead *et al.* [33] using MUSCLE. The 177 bp TBR group gathers the sequences from the Tbb177 and Tbg177 TBR sequence sets, while the 176 bp TBR group gathers the Tbbr176 and Tbg176 TBR sequence sets.

## Novel *Trypanozoon* qPCRs

All primers and probes for *Trypanozoon* detection are summarized in Table 1. We used IDT PrimerQuest to design a hydrolysis probe based qPCR for the *Trypanozoon* specific single-copy *GPI-PLC* gene (qGPI-PLC). The conventional M18S II PCR, as described in Deborggraeve *et al.* in [15], complemented with a hydrolysis probe for use in qPCR, as described by Bendofil *et al.* in [41], targets the multi-copy 18S rRNA of *Trypanozoon* and was abbreviated as q18S throughout this manuscript. IDT PrimerQuest was used to design a conventional (c177) and quantitative ($q177_D$) PCR based on the 80% consensus sequence of the Tbg177 set, aiming to target the 177 bp TBR group. IDT PrimerQuest was also used to generate a primer set for conventional PCR on the 80% consensus sequence of the Tbg176 set, called c176, with the aim to target the 176 bp TBR group. To amplify target sequences of both the 176 bp and 177 bp TBR groups in a multiplexed reaction, we used AlleleID 7 (PREMIER Biosoft) to design a common forward primer (qTBR-F), a 177-bp TBR group specific reverse primer and probe ($q177_T$), and a 176-bp TBR group specific reverse primer and probe ($q176_T$) based on the consensus sequences of the Tbg177 and Tbg176 sets. The position of the primers and probes targeting TBR are shown in S1 Fig.

## Conventional PCR

Conventional PCR was performed in a Biometra T3 using HotStarTaq Plus (Qiagen). Amplification was performed in 1x Coral Load Buffer, using 500 nM of each forward and reverse primer (IDT), 200 nM of each nucleotide (Eurogentec), 25 mM $MgCl_2$ and 2 μl pure parasite

**Table 1. Primers and probes for PCR and qPCR detection of target sequences of GPI-PLC, 18S rDNA, and the 176 bp and 177 bp TBR groups in *Trypanozoon*.**

| PCR | Oligo | Sequence | Length (bp) |
|---|---|---|---|
| qGPI-PLC | qGPI-PLC-F | CCCACAACCGTCTCTTTAACC | 106 |
| | qGPI-PLC-R | GGAGTCGTGCATAAGGGTATTC | |
| | qGPI-PLC-P | FAM-ACACCACTTTGTAACCTCTGGCAGT-MGB | |
| q18S | M18S II-F | CGTAGTTGAACTGTGGGCCACGT | 150 |
| | M18S II-R | ATGCATGACATGCGTGAAAGTGAG | |
| | q18S-P | VIC-TCGGACGTGTTTTGACCCACGC-MGB | |
| c177/q177$_D$ | c177-F | GCAACAAAGCTATTTAATGGTCCT | 109 |
| | c177-R | GCACACTTGTAATTAATATGGCACA | |
| | q177$_D$-P | FAM-TGCGCAGTTAACGCTATTATACACA-MGB | |
| c176 | c176-F | GTGCAACAAAGCTAATAAATGGTTC | 165 |
| | c176-R | TAAAGAACAGCGTTGCAAACTT | |
| q177$_T$ | qTBR-F | CGCAGTTAACGCTATTATACA | |
| | q177$_T$-R | GGACCATTAAATAGCTTTGTTG | 152 |
| | q177$_T$-P | NED-TGCCATATTAATTACAAGTGTGC-MGB | |
| q176$_T$ | qTBR-F | CGCAGTTAACGCTATTATACA | |
| | q176$_T$-R | GAACCATTTATTAGCTTTGTTG | 151 |
| | q176$_T$-P | FAM-TGCAACGCTGTTCT-MGB | |

DNA (10 ng/µl) in a 20 µl reaction. PCR cycling consisted of 95˚C for 5 min, followed by 29 cycles of 94˚C for 30 seconds, 60˚C for 30 seconds and 72˚C for 30 seconds for c177, c176, and M18S II [15]. Conventional PCR using TBR primers described in Masiga *et al.* [27], Mumba *et al.* [30] and Becker *et al.* [29] was performed at annealing temperatures of 55˚C, 60˚C and 66˚C respectively. After a final extension of 5 minutes at 72˚C, amplification reactions (10 µl) were visualized on 2% agarose after 135V for 30 minutes and stained in 0.5 mg/ml ethidium bromide. A GeneRuler 100bp plus DNA ladder (Thermo Scientific) was used for amplicon size estimation. Reactions were considered positive if bands of the expected length were observed. Semi-quantitative conventional PCRs were performed by using 7 serial fivefold dilutions: 1000 fg/µl, 200 fg/µl, 40 fg/µl, 8 fg/µl, 1.6 fg/µl, 0.32 fg/µl and 0.064 fg/µl of pure parasite DNA of each *Trypanozoon* strain.

## Quantitative PCR and RCN calculations

qPCR amplification was performed in a Quantstudio 5 (Applied Biosystems) using 1x PerfeCTa qPCR Toughmix (Quantabio), 300 nM of each forward and reverse primer (IDT), 100 nM probe (Thermo Scientific) and 5 µl of pure parasite DNA in a total volume of 20 µl. qPCR cycling consisted of 45˚C for 5 minutes, 95˚C for 10 minutes, followed by 35 cycles of 95˚C for 15 seconds and 60˚C for one minute. Analytical sensitivity and qPCR efficiency were calculated, using serial tenfold dilutions of pure parasite DNA: 100 pg/µl, 10 pg/µl, 1 pg/µl, 100 fg/µl, 10 fg/µl and 1 fg/µl of two *T.b.g.* clones, LiTat 1.6 and AnTat 11.17. In duplexed qPCR reactions, FAM-labelled probes for qGPI-PLC, q177$_D$ or the qPCR described by Mumba *et al.* ([30]), here abbreviated as qM, were combined with a VIC-labelled q18S probe to allow the calculation of relative copy-numbers (RCN). These RCN, were calculated by subtracting the $C_q$-values obtained in qGPI-PLC, q177$_D$ or qM from the $C_q$-value in q18S, resulting in a $\Delta C_q$-value that was transformed to $2^{-\Delta Cq}$ and averaged between replicates to yield the RCNs for each of these targets. The qTBR reaction is performed as a triplex qPCR reaction with a NED-labelled q177$_T$ probe, a FAM-labelled q176$_T$ probe and a VIC-labelled q18S probe. Here, RCNs were calculated by subtracting the $C_q$-values obtained in q177$_T$ or q176$_T$ from the $C_q$-

value in q18S, resulting in a $\Delta C_q$-value that was transformed to $2^{-\Delta Cq}$ and averaged between replicates to yield the RCNs for each TBR target. Graphical analysis was performed using R (3.5.2) in RStudio (1.1.463) with packages "ggplot" (3.2.1), "ggrepel" (0.8.1) and "viridis" (0.5.1).

## Results

### Novel TBR sequences identified by BLAST reveal the existence of two TBR groups

BLAST analysis of the sole published TBR-sequence (K00392.1) resulted in 99 hits on *T.b.b.* and 1 hit on *T.b.r.* in the *T. brucei* database, and 100 hits in the *T.b.g.* database. All hits were tandem repeats of two to four TBR sequences with an average size of 552-bp in the *T.b.* database and 761-bp in the *T.b.g.* database (S1 File). Alignment of *HhaI* extracted individual TBR-repeats revealed that none were 100% identical to K00392.1 (S1 File). In addition to single nucleotide polymorphisms (SNPs) and indels, also larger inserts and deletions were seen, as previously reported by others, but these were not numerous enough for further analysis [20, 33, 34]. Remarkably, a few key SNPs permitted to sort these individual TBR sequences into two major groups that were either 177 bp or 176 bp long. The 99 *T.b.b.* hits, contained 245 individual TBR sequences that formed the Tbb177 TBR sequence set. The 100 *T.b.g.* hits contained 153 sequences that formed the Tbg177 TBR sequence set, and 141 sequences that formed the Tbg176 set. Three individual TBR sequences obtained from the single *T.b.r.* hit were joined with two 176 bp sequences from the *T.b.b.* database, to form the Tbbr176 sequence set. For each of these four TBR sequence sets, 80% consensus sequences were generated and aligned against K00392.1 and two previously reported TBR variants, 177-T1 and 177-T2 [20, 33, 34] (Fig 1, S1 File). The Tbb177 and Tbg177 TBR sequence sets, together with the 177-T2 sequence, can be gathered into a larger 177 bp TBR group, while the Tbbr176 and Tbg176 TBR sequences sets share an indel and may be assembled into a broader 176 bp TBR group. Both TBR groups differed in a few key SNPs with K00392.1, yet they differed even more with each other.

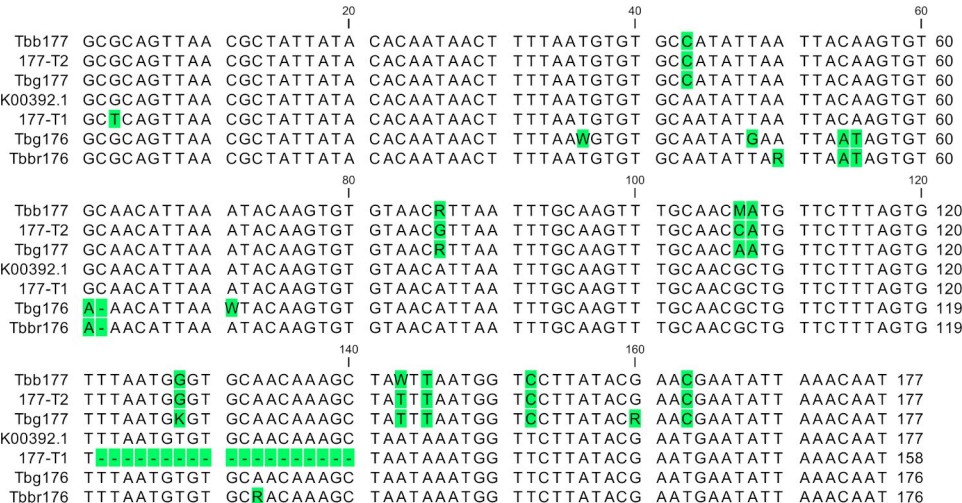

**Fig 1. Novel TBR groups identified by BLAST.** Alignment of the TBR sequence [K00392.1], both TBR variants, 177-T1 and 177-T2, and the 80% consensus sequences derived from each of the four TBR sequence sets. Polymorphisms in comparison to K00392.1 are indicated in green.

## Existing TBR PCRs are biased towards amplification of the 176-bp TBR group

For almost 40 years, the K00392.1 TBR sequence was the only template available for PCR primer design. All 6 published TBR primer sets [27–32] match 100% with the Tbg176 and Tbbr176 consensus sequences from the 176 bp TBR group (S2 Table). In contrast, most primer sets contain one or several mismatches in at least one of the primer binding regions for the Tbb177 and Tbg177 consensus sequences from the 177 bp TBR group (S2 Table). We developed a new primer set for conventional PCR, c177, to detect target sequences belonging to the 177 bp TBR group specifically, by intendedly mismatching both primers against the 176-bp TBR group sequences. Similarly, we developed a new primer set for conventional PCR to detect 176 bp TBR group, c176, that mismatches against the 177 bp TBR group sequences (S2 Table). We compared the analytical sensitivity in conventional PCR of c177 and c176 with the TBR PCR previously published by Masiga *et al.* [27], and to conventional PCR adapted versions of the TBR PCRs published by Becker *et al.* [29] and Mumba *et al.* [30]. Amplification of the 18S rDNA, using M18S II, as described in Deborggraeve *et al.* [15], was used as an external standard for detection of *Trypanozoon* DNA. In total, we tested 36 different *Trypanozoon* strains: 8 *T.b.b.*, 10 *T.b.g.* I, 2 *T.b.g.* II, 9 *T.b.r.*, 3 *T. ev.* A, 2 *T. ev.* B, 1 *T.eq.* B and 1 *T.eq.* O, in semi-quantitative conventional PCR (S1 Table). None of the conventional TBR PCRs resulted in the specific detection of a particular taxon, yet, not all TBR PCRs detected *Trypanozoon* DNA with the same sensitivity (S2 Fig). For example, c177 outperformed all other TBR PCR sets in *T.b.g.* I detection, while despite comparable DNA content in each dilution series according to M18S II, some PCRs that target the 176 bp TBR group were not successful in amplifying some of the *T.b.g.* I strains (Fig 2). Conventional PCR on TBR suffers from the difficult interpretation of the electrophoretic patterns possibly causing confusion about the specificity of the PCR. While the main amplicon of each TBR PCR had the expected length, many larger-sized bands, including bands that were the size of one full TBR repeat larger than the main amplicon occurred at higher amounts of template DNA.

## A multiplex qPCR to ameliorate amplification and copy-number calculation for target sequences of each of both TBR groups

We designed a multiplex qPCR, consisting of a common forward primer, qTBR-F, and specific reverse primers, q177$_T$-R and q176$_T$-R, and probes, q177$_T$-P and q176$_T$-P, for simultaneous amplification of target sequences of both the 177 bp and the 176 bp TBR group. When combined with 18S rDNA qPCR amplification, this novel qPCR triplex, called qTBR, allows accurate RCN calculation for each TBR target sequence using q18S as internal standard. We compared the qTBR with two other TBR qPCRs: the qPCR described by Mumba and co-workers [30], here, abbreviated as qM, and the q177$_D$, using the primers of the conventional TBR PCR, c177, complemented with a probe for qPCR, q177$_D$-P. Combining q18S in a duplex qPCR with qM or q177$_D$ allows to calculate the RCNs for these respective target TBR sequences, while the RCNs of 18S rDNA were calculated using a duplex qPCR containing qGPI-PLC and q18S. All novel qPCRs were first tested on a dilution series of DNA on two *T.b. g.* type I strains. For both strains, the target sequences for the 177 bp group, q177$_D$ and q177$_T$, could detect 5 fg of DNA, while $C_q$-values varied little between replicates of the same dilutions in simplex, duplex or triplex format (S3 Fig). In addition, qPCR efficiency had acceptable slopes between -3.1 and -3.6 in most formats (S4 Fig). Next, we tested gDNA from 77 *Trypanozoon* strains and clones: 12 *T.b.b.*, 35 *T.b.g.* I, 2 *T.b.g.* II, 11 *T.b.r.*, 21 *T. ev./eq.* A, 3 *T. ev.* B, 1 *T. eq.* B and 4 *T. eq.* O (S1 Table). We found that the RCNs of q18S showed little variation within and between the different *Trypanozoon* taxa. In contrast, the RCNs of the TBR repertoire

**Fig 2. Semi-quantitative conventional TBR PCRs on *Trypanozoon*.** Semi-quantitative conventional TBR PCRs using the c177, c176, Masiga, Becker, Mumba, and M18S II primer sets on fivefold serial dilutions containing 2000, 400, 80, 16, 3.2, 0.64, 0.128 or 0 fg of pure genomic *Trypanozoon* DNA. Each of the six gels shows the electrophoretic results of one conventional PCR tested on 3 *T.b.r.* (upper part of the gels) and 3 *T.b.g.* (lower part of the gels) strains, separated by 5 μl of the Generuler 100-bp DNA ladder (Thermo Scientific).

varied greatly between and even within the different *Trypanozoon* taxa (Fig 3). The 177-bp TBR group, assayed using q177$_D$ and q177$_T$ had RCNs ranging from tens to hundredths for most *Trypanozoon*, including *T.b.g.* I, and up to thousands in the case of *T.ev./eq.* A. In

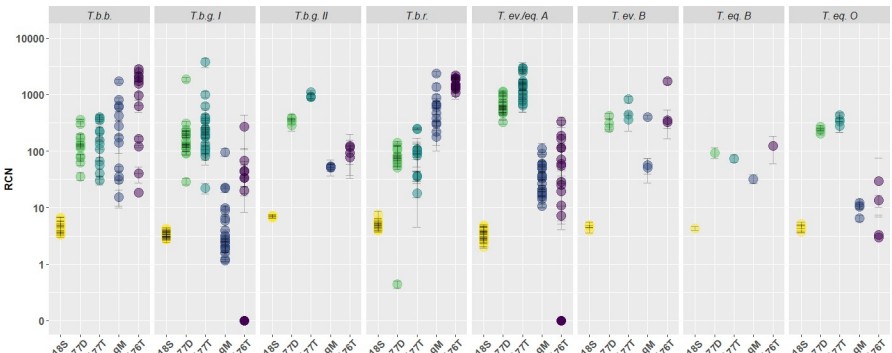

**Fig 3. RCNs for 18S rDNA and the TBR repertoire within 77 *Trypanozoon* strains.** All *Trypanozoon* strains were tested at 50 ng of pure genomic DNA. Mean and standard deviation were calculated using three replicates for each sample. RCNs for 18S rDNA were calculated using the $\Delta C_q$-method between q18S and qGPI-PLC. RCNs for the TBR targets q177$_D$, q177$_T$, qM and q176$_T$ were calculated using the $\Delta C_q$-method with q18S as reference.

contrast, in qM, RCNs of the target sequence ranged from thousands in *T.b.r*, to fewer than 10 in most *T.b.g.* I strains. Similarly, the $q176_T$ target sequence RCNs ranged from thousands in *T.b.b.* and *T.b.r.* to undetectably low in 15 out of 35 *T.b.g.* I and 10 out of 23 *T. ev./eq.* A strains.

## The qTBR can be used to assign *Trypanozoon* strains into TBR genotypes

By plotting the target sequence RCNs obtained for $q177_T$ against those obtained for $q176_T$, the 77 *Trypanozoon* strains present in this collection are scattered along a continuum whereby both ends can be named after the geographical region from where most strains were isolated (Fig 4). The first TBR genotype, called TBR-East, is here defined as having at least 1.2-fold higher RCNs observed in $q176_T$ than in $q177_T$ and consists of *T.b.r.* and *T.b.b.* strains that mostly originate from East Africa. All the *T.b.r.* strains were isolated in Uganda, Rwanda or Kenya, except TRPZ 210, which was isolated in Zambia. Six out of nine *T.b.b.* strains belong to this TBR-East genotype. Five of them originated from East Africa (Uganda, Kenya and Tanzania), yet AnTat 17.1 was isolated from a sheep in Kongo-Central province of The Democratic Republic of the Congo (DRC). The second TBR genotype, called TBR-West, is here defined as having higher or equal RCNs for the target sequence in $q177_T$ than in $q176_T$. It comprises representatives of all non-*T.b.r.* species and subspecies, including historical *T.b.g.* I strains from Côte d'Ivoire, but also the most recent *T.b.g.* I strain in our collection, i.e. MM01, isolated in 2008 in Kwilu province in DRC. The *T.b.g.* II strains ABBA and LIGO and the remaining *T.b.b.* strains were isolated in West Africa (Côte d'Ivoire, The Gambia, Nigeria), except *T.b.b.* J10, which was isolated from a hyena in Zambia. All taxa of dyskinetoplastic trypanosomes are represented in this TBR-West genotype: 13 out of 23 *T.ev./eq.* A (China, Ethiopia, Kenya, Morocco, unknown origin) and all four *T. eq.* O strains (RSA, Ethiopia, Venezuela), although the three *T.ev.* B (Ethiopia and Kenya) and the single *T.eq.* B (Morocco) are positioned in the middle between both TBR genotypes. A third TBR genotype, called TBR-Central, comprises strains that have detectable RCN for $q177_T$, yet remain negative for $q176_T$. This genotype corresponds to *T.b.g.* I strains isolated mainly in Cameroon and in the East-Kasaï province in DRC, but also holds 10 out of the 23 *T.ev./eq.* A strains (Brazil, Colombia, Indonesia, Kazakhstan, the Philippines, South-America, unknown origin).

## Discussion

Over the last 40 years, TBR PCRs have increasingly been used for detecting *Trypanozoon* infections in vertebrates and insects. During this period, primer sets were often repositioned in different PCR formats, yet all those positional adaptations were based on the single TBR sequence reported by Sloof *et al.* [25]. Here, we have shown that this TBR sequence is more heterogenous than previously assumed. The rediscovered 177 bp TBR group allows to ameliorate amplification of most non-*rhodesiense Trypanozoon*, which is in particular relevant for *T.b.g.* I and the dyskinetoplastic trypanosomes. Some strains belonging to these taxa are not detected by primer sets that solely target the 176 bp TBR group. The dual TBR probes in the qTBR improve amplification of all Trypanozoon taxa, which is particularly relevant for trypanosomiasis infections caused by *T.b.g* I in insects and mammals [42]. Plotting the target sequence RCNs of $q177_T$ versus those of the $q176_T$ classifies *Trypanozoon* strains into two opposing genotypes: TBR-East and TBR-West, whose names roughly refer to the geographical origin of the strains making up these opposites. This East-West dichotomy in *Trypanozoon* has been observed earlier via various genotyping techniques such as zymodemes, VSG repertoire, microsatellites, and even genome-wide SNP analysis [11, 23, 39, 40, 43, 44]. These latter techniques have some disadvantages such as the requirement of high amounts of input

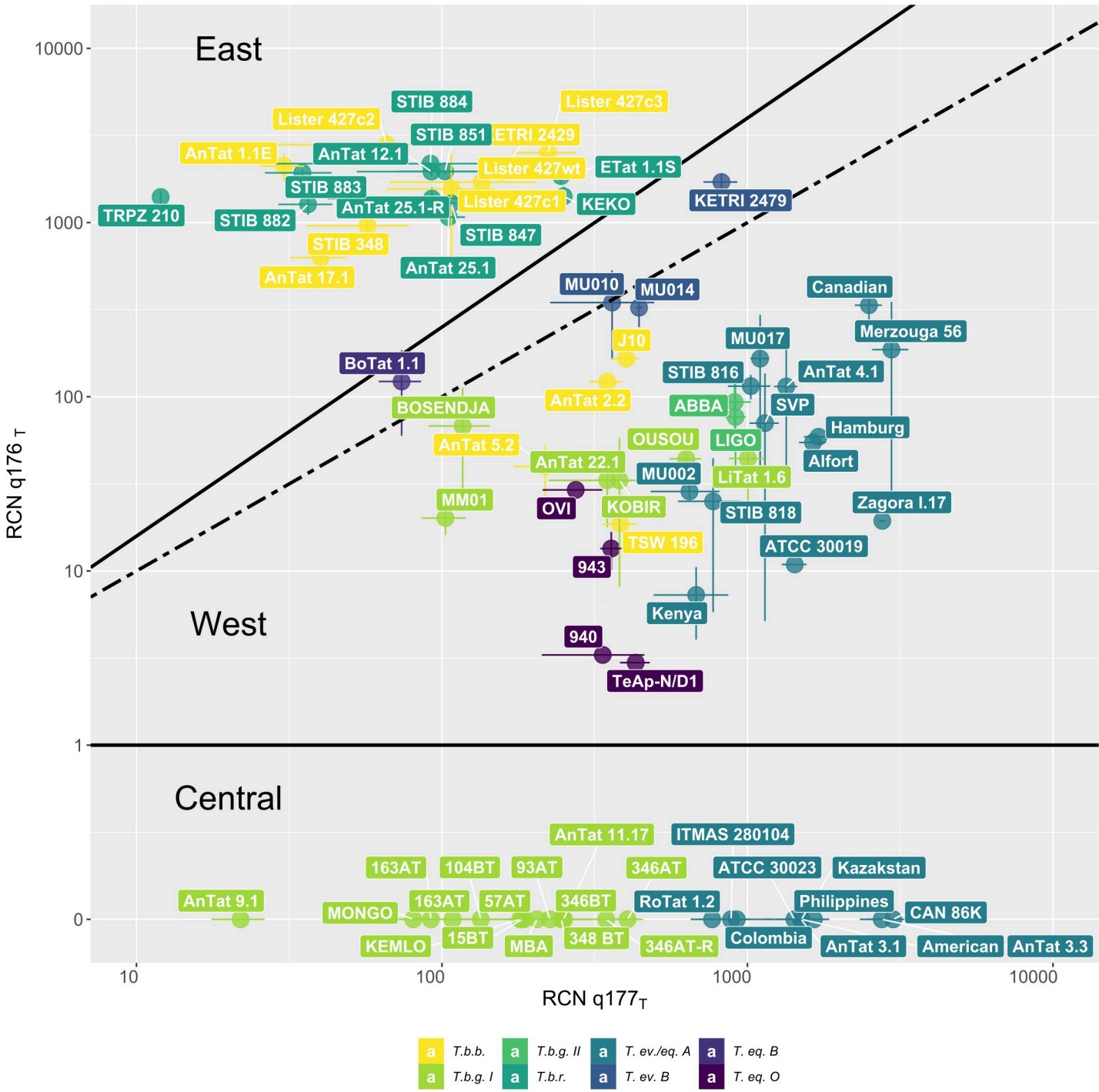

**Fig 4. Three TBR genotypes among 77 *Trypanozoon* strains.** All *Trypanozoon* stocks were tested at 50 ng of pure genomic DNA using the qTBR. Mean and standard deviation were calculated using three replicates for each sample. RCNs for target sequences $q177_T$ and $q176_T$ were calculated using the $\Delta C_q$-method with q18S as reference and plotted in a scatterplot. Two solid lines were drawn to divide the *Trypanozoon* strains into roughly 3 TBR-genotypes: East, West and Central. The line at y = 1 separates TBR-Central from both TBR-West and TBR-East, while the line at y = 1.2x separates TBR-East from TBR-West and TBR-Central. The dotted line at y = x represents equal RCNs for $q177_T$ and $q176_T$.

material, multiple PCR reactions, difficult interpretation of banding patterns or requiring bio-informatic analysis. In contrast, the qTBR allows a rough genotyping of strains by one single multiplex qPCR. Because TBR sequences form the central core of MCs, absence of amplification may therefore indicate loss of certain MC sets. According to this view, the TBR-Central genotype may just appear as a degenerate form of the TBR-West genotype, composed of strains that have lost MCs that contain 176 bp TBR sequences. The qTBR is one of the few techniques that is able to demonstrate such microheterogeneity within both *T.b.g.* I and *T. evansi*, two species assumed to be exclusively clonally propagated [10, 45]. *T.b.g.* I strains iso-lated in Kwilu and Kasaï-Oriental provinces in the DRC suggest that different TBR genotypes are circulating in foci separated less than 1000 km from each other. The 177 bp TBR group is detectable in all *T.b.g.* I strains present in this collection. However, more strains from East Africa and Central Africa ought to be included for a better geographical coverage of *T.b.g.* I [46, 47]. In addition, another limit of our study is the absence of recently isolated *T.b.b.* strains from West Africa and Central Africa in our *Trypanozoon* collection. Trypanosomes causing HAT can be found among all three TBR genotypes and cannot unequivocally be differentiated from those causing AAT or NTTAT. Nevertheless, presence of *Trypanozoon* DNA in human clinical samples should always warrant special attention, since atypical, *e.g.* with *T.b.g.* II, or even opportunistic *Trypanozoon* infections, e.g. with *T.b.b.* and *T. ev.*, are known to occur [8, 48]. With the limited TBR sequence information available today, we cannot exclude that more TBR groups may be discovered within larger and more diverse collections of *Trypanozoon*, perhaps even specific for certain *Trypanozoon* taxa. Sequencing the *Trypanozoon* repetitive DNA, preferentially using sequencing platforms that overcome the limitations imposed by tan-dem repeat sequences, will be crucial to further understand the evolution of MC and the diver-sity in TBR content of African trypanosomes [49].

## Supporting information

**S1 Fig. Position of the primers and probes targeting TBR.** A representation of a TBR sequence as tandem repeat (yellow) showing the relative position of primers and probes used in conventional and quantitative PCR. Arrows indicate the 5'– 3' direction of primers, while for probes, circles indicate fluorophores and diamonds indicate quenchers. In A, blue indicates the c177 and the $q177^D$ set, while red represents the c176 set. In B, the green arrow represents the common primer qTBR-F, while the blue primer and probe indicate the $q177_T$ set, and the red primer and probe indicate the $q176^T$ set.
(TIF)

**S2 Fig. Conventional TBR PCR on 36 *Trypanozoon* strains.** Semi-quantitative conventional PCRs using the Masiga, Becker, Mumba, c177, c176 and M18S II primer sets on fivefold serial dilutions containing 2000, 400, 80, 16, 3.2, 0.64, 0.128 or 0 fg of pure parasite DNA per lane. In total, 36 *Trypanozoon* strains were tested. Each of the gels shows the electrophoretic results of one of the conventional PCRs tested on 6 *Trypanozoon* strains (3 above and 3 below), sepa-rated by 5 μl of the Generuler 100-bp DNA ladder (Thermo Scientific).
(TIF)

**S3 Fig. qPCR efficiency of novel qPCRs in different multiplex formats.** $C^q$-values obtained from a tenfold dilution series from 500 pg down to 5 fg of pure parasite DNA (in elution buffer) of two *T.b.g.* I clones: AnTat 11.17 and LiTat 1.6 in different qPCR formats: simplex, duplex (in combination with q18S) or triplex (as qTBR). The slope of qPCR efficiency was esti-mated by fitting a linear trendline on $C^q$ values and the log transformed concentrations.
(TIF)

**S4 Fig. Analytical sensitivity of novel qPCRs in different multiplex formats.** $C_q$-values obtained from a tenfold dilution series from 500 pg down to 5 fg of pure parasite DNA (in elution buffer) of two *T.b.g.* I clones: AnTat 11.17 and LiTat 1.6 in different qPCR formats. Each of the novel qPCRs was first tested individually in simplex format (A). qGPI-PLC, q177$^D$ or qM, was combined with q18S in duplex format (B). q177$^T$ and q176$^T$ were combined with q18S in triplex format, representing the qTBR (C).
(TIF)

**S1 File. Extracted TBR sequences and arrangement into TBR sets.** Tandem TBR sequences obtained by running the TBR-sequence [K00392.1] on the Trypanosoma Blast Server using BLASTn on the Sanger Institute website against the *T.b.b.* and *T.b.g.* database (sheets Hits T.b. and Hits T.b.g.), individual TBR repeats extracted using *HhaI* (sheets HhaI T.b. and HhaI T.b.g.) and TBR sequence sets, sorted per key SNPs and subspecies (sheets Tbb177, Tbg177, Tbg176 and Tbbr176). The 80% consensus sequences for each TBR sequence set are described on the last line of these sheets.
(XLSX)

**S1 Raw images. Raw images of Fig 2.**
(PDF)

**S2 Raw images. Raw images of S2 Fig.**
(PDF)

**S1 Table. *Trypanozoon* strains and presumed taxon.** Collection of *Trypanozoon* strains with taxon, strain name, clone status, host, country, area and year of isolation and the results of the PCR typing of this strains using TgsGP, SRA, RoTat 1.2, EVA B, MORF2-REP and their TBR genotype according to qTBR.
(XLSX)

**S2 Table. Existing TBR PCR sets and compatibility with the novel TBR sequence sets.** Existing TBR PCR sets with literature reference, type of PCR, primer names and sequence, matching TBR sets and mismatching TBR sets.
(XLSX)

## Acknowledgments

We would like to thank Jeroen Swiers (ITM, Antwerp) for excellent assistance in cryobiology.

## Author Contributions

**Conceptualization:** Nick Van Reet, Philippe Büscher.

**Data curation:** Nick Van Reet.

**Formal analysis:** Nick Van Reet.

**Funding acquisition:** Nick Van Reet, Philippe Büscher.

**Investigation:** Nick Van Reet, Sara Dehou, Nicolas Bebronne.

**Methodology:** Nick Van Reet, Stijn Deborggraeve.

**Project administration:** Nick Van Reet, Sara Dehou, Nicolas Bebronne.

**Resources:** Pati Patient Pyana, Nicolas Bebronne, Stijn Deborggraeve, Philippe Büscher.

**Supervision:** Nick Van Reet, Philippe Büscher.

**Validation:** Sara Dehou, Nicolas Bebronne.

**Visualization:** Nick Van Reet.

**Writing – original draft:** Nick Van Reet, Philippe Büscher.

**Writing – review & editing:** Nick Van Reet, Pati Patient Pyana, Sara Dehou, Nicolas Bebronne, Stijn Deborggraeve, Philippe Büscher.

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
