## [Decision Letter · Decision Letter 0]

28 Jul 2021

PONE-D-21-19109

Single nucleotide polymorphisms and copy-number variations in the Trypanosoma brucei repeat (TBR) sequence can be used to enhance amplification and genotyping of Trypanozoon strains

PLOS ONE

Dear Dr. Van Reet,

Thank you for submitting your manuscript to PLOS ONE. After careful consideration, we feel that it has merit but does not fully meet PLOS ONE’s publication criteria as it currently stands. Therefore, we invite you to submit a revised version of the manuscript that addresses the points raised during the review process.

We look forward to receiving your revised manuscript.

Kind regards,

Maria Stefania Latrofa

Academic Editor

PLOS ONE

Journal Requirements:

Reviewers' comments:

Reviewer's Responses to Questions

**Comments to the Author**

1. Is the manuscript technically sound, and do the data support the conclusions?

Reviewer #1: Yes

Reviewer #2: Yes

2. Has the statistical analysis been performed appropriately and rigorously? 

Reviewer #1: Yes

Reviewer #2: Yes

3. Have the authors made all data underlying the findings in their manuscript fully available?

Reviewer #1: Yes

Reviewer #2: Yes

4. Is the manuscript presented in an intelligible fashion and written in standard English?

Reviewer #1: Yes

Reviewer #2: Yes

5. Review Comments to the Author

Reviewer #1: Review of Single nucleotide polymorphisms and copy-number variations in the Trypanosoma brucei repeat (TBR) sequence can be used to enhance amplification and genotyping of Trypanozoon strains by Van Reet et al.

In this paper Van Reet and co-authors analysed genetic variation in TBR sequences and developed a novel quantitative PCR molecular test (qTBR) which allows to identify Trypanozoon taxa.

The study is technically sound, and the manuscript is well written. Molecular detection of Trypanosoma is an important approach for the early diagnosis of disease. In my opinion, the manuscript represents an incremental advance that is of interest to the field.

Only minor issues should be addressed.

Line 56 and 81-86: Update the reference (ref 1; ref 20-23)

Lines 57-59: put the HAT in 58 West and Central Africa into wider context than 2018

Lines 95, 96 and throughout the paper: About the possibility to assess the geographical origin of Trypanosoma strains using the TBR-PCR, more caution should be used. The future analysis of a higher number of strains would be useful to support this finding.

Reviewer #2: Comments to “Single nucleotide polymorphisms and copy-number variations in the Trypanosoma brucei repeat (TBR) sequence can be used to enhance amplification and genotyping of Trypanozoon strains” by Van Reet et al.

Van Reet et al proposes an interesting work on a molecular marker (TBR) used for decades in the field of Trypanosoma diagnostics. As rightly noticed by the authors, very few knowledge on this marker is available. Thus, the idea of this study is to give more insight on the molecular properties of this marker taking advantage of sequences available on the databases and of a large panel of Trypanosoma spp DNA from different geographical origin.

To do so, they first looked for TBR reads available from Tbb and Tbg database and aligned and sorted them thanks to the presence of SNPs. From this first analysis, they observed two groups of sequences: one of 177bp and the other of 176bp; both group differing from few SNPs but that cannot discriminate Tbb from Tbg. Variation of the copy number of repeats have been also detected.

Interestingly, the authors then observed that most of the current TBR primers preferentially matched with the 176bp TBR group and thus, are potentially missing the 177bp group information. They decided to design new primer sets for conventional PCR targeting both groups (calling PCR c177 and c176) and to perform semi-quantitative PCRs to compare sensitivity of these new primers with the previous ones. As suspected by in silico analysis, 177bp primer sets allowed the amplification of some Tbg1 strains that did not give PCR signal when amplified by c176 or the classical published primers.

Consequently, the authors design a multiplex qPCR with specific reverse primers for 176 and 177 (called q177T-R and q176T-R) altogether with q177T and q176T probes. They combined these 2 PCR with a 18S rDNA qPCR that will allow to normalize and calculated the relative copy-number (RCN) of each TBR target. They showed that the TBR repertoire is important within and between the different Trypanozoon taxa, with TBR RCN ranging from around 10 to more than 1000 in some cases.

To explore more deeply this variability, Van Reet et al decided to analyse the data set by plotting q177t against q176t for 77 Trypanozoon strains coming from different origins (East/Central/West Africa). Three clusters, corresponding to the three geographic origins were highlighted, independently of the Trypanosoma species.

General Comments:

Despite the complexity of the methodology used, it is clear that the authors did a great effort to make the paper as clear as possible. They go straightforward even sometimes, it makes difficult to catch the ideas from the first read. From a technical point of view, the experiments are well designed and the controls are properly used.

On one hand, the new data generated on TBR marker are interesting from a conceptual/fundamental point of view. For this, the MS deserves to be published. On the other hand, the expectation of reader is a little disappointed because it is hard to imagine than such a tool could be used to improve the diagnosis, especially to discriminate the animal-infective trypanosomes from the human-infective ones. Currently, there is an urgent need to develop new tools to do so, and the approach proposed by Van Reet et al does not fill this gap.

Specific comments:

Please define “RCN” before the “Results” section. The same for “qM”.

Figure 4: lack of Tbb from Central Africa.

Figure 4: Improve the color code to make easier the recognition of the different species.

Table 1: Find a way to help the reader to understand the primer combinations between the conventional, duplex and triplex (q)PCR. It will facilitate the reading and understanding of the MS.

6. PLOS authors have the option to publish the peer review history of their article (what does this mean?). If published, this will include your full peer review and any attached files.

Reviewer #1: No

Reviewer #2: No

---

## [Author Response · Author response to Decision Letter 0]

27 Sep 2021

Reviewer #1: Review of Single nucleotide polymorphisms and copy-number variations in the Trypanosoma brucei repeat (TBR) sequence can be used to enhance amplification and genotyping of Trypanozoon strains by Van Reet et al.

In this paper Van Reet and co-authors analysed genetic variation in TBR sequences and developed a novel quantitative PCR molecular test (qTBR) which allows to identify Trypanozoon taxa.

The study is technically sound, and the manuscript is well written. Molecular detection of Trypanosoma is an important approach for the early diagnosis of disease. In my opinion, the manuscript represents an incremental advance that is of interest to the field.

Only minor issues should be addressed.

Line 56 and 81-86: Update the reference (ref 1; ref 20-23)

In line 56, we complemented the historical reference of Hoare with the more recent Radwanska et al 2018 “Salivarian Trypanosomosis: A Review of Parasites Involved, Their Global Distribution and Their Interaction With the Innate and Adaptive Mammalian Host Immune System”.

In line 81, we added a reference after the sentence. We updated the statement in line 84 with a reference on the genome of Trypanosoma brucei gambiense by Jackson et al 2010 “The Genome Sequence of Trypanosoma brucei gambiense, Causative Agent of Chronic Human African Trypanosomiasis”, and moved this sentence after the statement on other non-repetitive DNA which refers to T. brucei. In line 86, we omitted the original references 20, 21 and 22 and only retained ref 20. The original reference 23 (now reference 25) is not mentioned in lines 81-86.

This section [lines 80-87] now reads as “Around 100 MCs, sized 50-150 kb, are present in the nuclear DNA of T. brucei and they represent almost 10% of the nuclear genome (19). It is estimated that roughly 55% of each MC, and thus 5.5% of the nuclear DNA in T. brucei, consists of such TBR repeats (19,20). The non-repetitive DNA on MCs carries an important part of the silent VSG gene repertoire, with most MCs having complete VSG genes that can be transposed to the VSG expression site during the early stages of an infection (20). In T.b.g., the average lengths of the MCs are smaller, being 25 to 50 kb, and the estimated copy-numbers vary between a few to up to 100 (21–24).”

Lines 57-59: put the HAT in 58 West and Central Africa into wider context than 2018

We assume that the reviewer wants us to mention that gambiense is targeted for elimination by WHO. This section [lines 57-59] now reads as “T.b. gambiense (T.b.g.) is responsible for chronic human African trypanosomiasis (HAT), a disease targeted for elimination by the World Health Organization that still accounted for 977 patients reported in West and Central Africa in 2018 (3). Annually, less than 100 cases are due to T.b. rhodesiense (T.b.r.), which causes acute HAT in East Africa (4).” 

Alternatively, we could give the exact numbers of reported cases for 2020 for both gambiense and rhodesiense by referring to https://www.who.int/data/gho/data/indicators/indicator-details/GHO/hat-tb-gambiense and https://www.who.int/data/gho/data/indicators/indicator-details/GHO/number-of-new-reported-cases-of-human-african-trypanosomiasis-(t-b-rhodesiense)

Lines 95, 96 and throughout the paper: About the possibility to assess the geographical origin of Trypanosoma strains using the TBR-PCR, more caution should be used. The future analysis of a higher number of strains would be useful to support this finding.

We added more caution by adjusting the following statements relating to the possibility to assess the geographical origin using TBR-PCR.

Line 51, we replaced ‘it is possible to asses’ to ‘may hint at’. This section [lines 49-51] now reads as

“In addition, variations in the TBR content of Trypanozoon are apparently so large that a given qTBR profile may hint at the geographical origin of a given strain.”

Line 95, we replaced “asses” to “may suggest”. This section [lines 94-97] now reads as

“Furthermore, we show that single nucleotide polymorphisms and copy-number variations in the TBR sequences can be exploited to improve the amplification of all Trypanozoon taxa using a newly developed quantitative TBR-PCR, called qTBR, that may even allow to suggest the geographical origin of certain strains.”

Line 271, we added “present in this collection”. This section [lines 274-277] now reads as

“By plotting the target sequence RCNs obtained for q177T against those obtained for q176T, the 77 Trypanozoon strains present in this collection are scattered along a continuum whereby both ends can be named after the geographical region from where most strains were isolated (Fig 4).“

Line 338, we added “within larger and more diverse collections of Trypanozoon” . This section [lines 340 - 342] now reads as 

“With the limited TBR sequence information available today, we cannot exclude that more TBR groups may be discovered within larger and more diverse collections of Trypanozoon, perhaps even specific for certain Trypanozoon taxa.”

 

Reviewer #2: Comments to “Single nucleotide polymorphisms and copy-number variations in the Trypanosoma brucei repeat (TBR) sequence can be used to enhance amplification and genotyping of Trypanozoon strains” by Van Reet et al.

Van Reet et al proposes an interesting work on a molecular marker (TBR) used for decades in the field of Trypanosoma diagnostics. As rightly noticed by the authors, very few knowledge on this marker is available. Thus, the idea of this study is to give more insight on the molecular properties of this marker taking advantage of sequences available on the databases and of a large panel of Trypanosoma spp DNA from different geographical origin.

To do so, they first looked for TBR reads available from Tbb and Tbg database and aligned and sorted them thanks to the presence of SNPs. From this first analysis, they observed two groups of sequences: one of 177bp and the other of 176bp; both group differing from few SNPs but that cannot discriminate Tbb from Tbg. Variation of the copy number of repeats have been also detected.

Interestingly, the authors then observed that most of the current TBR primers preferentially matched with the 176bp TBR group and thus, are potentially missing the 177bp group information. They decided to design new primer sets for conventional PCR targeting both groups (calling PCR c177 and c176) and to perform semi-quantitative PCRs to compare sensitivity of these new primers with the previous ones. As suspected by in silico analysis, 177bp primer sets allowed the amplification of some Tbg1 strains that did not give PCR signal when amplified by c176 or the classical published primers.

Consequently, the authors design a multiplex qPCR with specific reverse primers for 176 and 177 (called q177T-R and q176T-R) altogether with q177T and q176T probes. They combined these 2 PCR with a 18S rDNA qPCR that will allow to normalize and calculated the relative copy-number (RCN) of each TBR target. They showed that the TBR repertoire is important within and between the different Trypanozoon taxa, with TBR RCN ranging from around 10 to more than 1000 in some cases.

To explore more deeply this variability, Van Reet et al decided to analyse the data set by plotting q177t against q176t for 77 Trypanozoon strains coming from different origins (East/Central/West Africa). Three clusters, corresponding to the three geographic origins were highlighted, independently of the Trypanosoma species.

General Comments:

Despite the complexity of the methodology used, it is clear that the authors did a great effort to make the paper as clear as possible. They go straightforward even sometimes, it makes difficult to catch the ideas from the first read. From a technical point of view, the experiments are well designed and the controls are properly used.

On one hand, the new data generated on TBR marker are interesting from a conceptual/fundamental point of view. For this, the MS deserves to be published. On the other hand, the expectation of reader is a little disappointed because it is hard to imagine than such a tool could be used to improve the diagnosis, especially to discriminate the animal-infective trypanosomes from the human-infective ones. Currently, there is an urgent need to develop new tools to do so, and the approach proposed by Van Reet et al does not fill this gap.

The reviewer is correct in stating that despite the improvements made in TBR PCR, we are not able to differentiate T.b. gambiense from non-gambiense. However, we firmly disagree that it would be hard to imagine that this PCR would improve diagnosis. Indeed, the possibility to diagnose a Trypanozoon infection caused by T.b. gambiense, is enormously improved considering the fact that previous TBR PCRs were not always able to amplify this parasite subspecies. A statement underlining this is moved higher up in the discussion to lines 314-316. “The dual TBR probes in the qTBR improve amplification of all Trypanozoon taxa, which is particularly relevant for trypanosomiasis infections caused by T.b.g I in insects and mammals (42).”

Specific comments:

Please define “RCN” before the “Results” section. The same for “qM”.

Done. Both abbreviations are now defined in the Material and methods section lines 169-172. 

Figure 4: lack of Tbb from Central Africa. 

The reviewer points to a lack of T.b.b. strains from Central Africa. However, this is not completely the case, T.b.b. AnTat 17.1 was isolated from a sheep in the Democratic Republic of the Congo in 1978 and is genotyped as TBR-East. However, we agree that the absence of recently isolated T.b.b. strains especially from DRC, is a limitation of this study. Therefore, we added this to the limits of the study and remarks on T.b.g. I already present in the discussion. This section [lines 333-336] now reads as 

“The 177 bp TBR group is detectable in all T.b.g. I strains present in this collection. However, more strains from East Africa and Central Africa ought to be included for a better geographical coverage of T.b.g. I (46,47). In addition, another limit of our study is the absence of recently isolated T.b.b. strains from West Africa and Central Africa in our Trypanozoon collection.”

Figure 4: Improve the color code to make easier the recognition of the different species.

We thank the reviewer for spotting this mistake. We omitted the “T.b.g. I MPXR” notation in Fig 4, which was a remnant of a previous version of this Figure. The updated Fig 4 still contains 8 colors (instead of 9) which cannot be further reduced. The current color scale (R-viridis) was specifically chosen to improve visualization for colorblindness and gray-scale printing. (https://cran.r-project.org/web/packages/viridis/vignettes/intro-to-viridis.html). 

Table 1: Find a way to help the reader to understand the primer combinations between the conventional, duplex and triplex (q)PCR. It will facilitate the reading and understanding of the MS.

We thank the reviewer for indicating that this is indeed difficult to understand from the current section. We agree that this section in Methods describing the primers and probes of Table1 could be improved. We tried to add more clarity by describing the targets in separate sentences. The first section now reads as [lines 130-135], while in line 139 we corrected the software version of AlleleID 7.

“We used IDT PrimerQuest to design a hydrolysis probe based qPCR for the Trypanozoon specific single-copy GPI-PLC gene (qGPI-PLC). The conventional M18S II PCR, as described in Deborggraeve et al in (15), complemented with a hydrolysis probe for use in qPCR, as described by Bendofil et al in (41), targets the multi-copy 18S rRNA of Trypanozoon and was abbreviated as q18S throughout this manuscript. IDT PrimerQuest was used to design a conventional (c177) and quantitative (q177D) PCR based on the 80% consensus sequence of the Tbg177 set, aiming to target the 177 bp TBR group.“ 

In the following section we added a sentence to introduce a supplementary figure to help visualize the difference between the conventional PCR and the duplex and triplex qPCR described in Table 1. [lines 143-144] “The position of the primers and probes targeting TBR are shown in S1_Fig.” Followed by a section [lines 491-497 ] in Supporting information. 

“S1_Fig. Position of the primers and probes targeting TBR

A representation of a TBR sequence as tandem repeat (yellow) showing the relative position of primers and probes used in conventional and quantitative PCR. Arrows indicate the 5’ – 3’ direction of primers, while for probes, circles indicate fluorophores and diamonds indicate quenchers. In A, blue indicates the c177 and the q177D set, while red represents the c176 set. In B, the green arrow represents the common primer qTBR-F, while the blue primer and probe indicate the q177T set, and the red primer and probe indicate the q176T set. “

Finally, to further improve the interpretation of duplex and triplex qPCR, we moved the explication of the duplex reaction before the triplex qTBR reaction. This section [lines 169-176] now reads as 

“In duplexed qPCR reactions, FAM-labelled probes for qGPI-PLC, q177D or the qPCR described by Mumba et al ((30)), here abbreviated as qM, were combined with a VIC-labelled q18S probe to allow the calculation of relative copy-numbers (RCN). These RCN, were calculated by subtracting the Cq-values obtained in qGPI-PLC, q177D or qM from the Cq-value in q18S, resulting in a �Cq-value that was transformed to 2-�Cq and averaged between replicates to yield the RCNs for each of these targets. The qTBR reaction is performed as a triplex qPCR reaction with a NED-labelled q177T probe, a FAM-labelled q176T probe and a VIC-labelled q18S probe. Here, RCNs were calculated by subtracting the Cq-values obtained in q177T or q176T from the Cq-value in q18S, resulting in a �Cq-value that was transformed to 2-�Cq and averaged between replicates to yield the RCNs for each TBR target”

---

## [Editor Report · Decision Letter 1]

5 Oct 2021

Single nucleotide polymorphisms and copy-number variations in the Trypanosoma brucei repeat (TBR) sequence can be used to enhance amplification and genotyping of Trypanozoon strains

PONE-D-21-19109R1

Dear Dr. Van Reet,

We’re pleased to inform you that your manuscript has been judged scientifically suitable for publication and will be formally accepted for publication once it meets all outstanding technical requirements.

Kind regards,

Maria Stefania Latrofa

Academic Editor

PLOS ONE

---

## [Editor Report · Acceptance letter]

15 Oct 2021

PONE-D-21-19109R1 

Single nucleotide polymorphisms and copy-number variations in the *Trypanosoma brucei* repeat (TBR) sequence can be used to enhance amplification and genotyping of *Trypanozoon* strains 

Dear Dr. Van Reet:

I'm pleased to inform you that your manuscript has been deemed suitable for publication in PLOS ONE. Congratulations! Your manuscript is now with our production department. 

Kind regards, 

on behalf of

Dr. Maria Stefania Latrofa 

Academic Editor

PLOS ONE